# Evidence for deleterious effects of immunological history in SARS-CoV-2

**Sanjana R. Sen**[1], **Emily C. Sanders**[2], **Alicia M. Santos**[2], **Keertna Bhuvan**[2], **Derek Y. Tang**[2], **Aidan A. Gelston**[2], **Brian M. Miller**[2], **Joni L. Ricks-Oddie**[3,4], **Gregory A. Weiss**[1,2,5]*

**1** Department of Molecular Biology & Biochemistry, University of California, Irvine, Irvine, CA, United States of America, **2** Department of Chemistry, University of California, Irvine, Irvine, CA, United States of America, **3** Center for Statistical Consulting, Department of Statistics, University of California, Irvine, Irvine CA, United States of America, **4** Biostatics, Epidemiology and Research Design Unit, Institute for Clinical and Translational Sciences, University of California, Irvine, Irvine, CA, United States of America, **5** Department of Pharmaceutical Sciences, University of California, Irvine, Irvine, CA, United States of America

* gweiss@uci.edu

**Data Availability Statement:** All relevant data are within the manuscript and its Supporting Information figures, tables, and file.

**Funding:** This study was funded by the UCI COVID-19 Basic, Translational and Clinical Research Fund

## Abstract

A previous report demonstrated the strong association between the presence of antibodies binding to an epitope region from SARS-CoV-2 nucleocapsid, termed Ep9, and COVID-19 disease severity. Patients with anti-Ep9 antibodies (Abs) had hallmarks of antigenic interference (AIN), including early IgG upregulation and cytokine-associated injury. Thus, the immunological memory of a prior infection was hypothesized to drive formation of suboptimal anti-Ep9 Abs in severe COVID-19 infections. This study identifies a putative primary antigen capable of stimulating production of cross-reactive, anti-Ep9 Abs. Binding assays with patient blood samples directly show cross-reactivity between Abs binding to Ep9 and only one bioinformatics-derived, homologous putative antigen, a sequence derived from the neuraminidase protein of H3N2 influenza A virus. This cross-reactive binding is highly influenza strain specific and sensitive to even single amino acid changes in epitope sequence. The neuraminidase protein is not present in the influenza vaccine, and the anti-Ep9 Abs likely resulted from the widespread influenza infection in 2014. Therefore, AIN from a previous infection could underlie some cases of COVID-19 disease severity.

## Introduction

Antigenic imprinting (AIM) or antigenic interference (AIN) occurs when the immune response adapted for a primary infection instead targets a similar, but not identical, pathogen [1]. While AIM describes imprinted immune responses procured in childhood, AIN occurs independent of age-cohort and can lead to an ineffective immune response due to antigenic drifts from primary infections or vaccines [2]. Since B-cells undergo affinity maturation after the primary infection, cross-reactive B-cells from previous infections can outcompete naïve Abs [3]. AIN ideally accelerates pathogen clearance by targeting highly conserved antigens; however, suboptimal targeting by non-neutralizing, Ab binding can exacerbate disease [3]. The range of outcomes observed in COVID-19, from asymptomatic to fatal, could result from a patient's immunological memory [1, 4].

(CRAFT), the Allergan Foundation, and the UCOP Emergency COVID-19 Research Seed Fund. The National Institute of General Medical Sciences also supported this paper through fellowship support to AMS (GM69337), the National Center for Advancing Translational Sciences supported JLR (TR001414), and the National Human Genome Research Institute awarded a grant to GAW (1R01HG009188-01).

**Competing interests:** The authors have declared that no competing interests exist.

Ab cross-reactivity from AIN causes a wide range of disease outcomes. For example, some Abs from healthy individuals previously exposed to other common human coronaviruses (hCoV) could cross-react with SARS-CoV-2 spike protein to neutralize viral pseudotypes [5]. However, other prepandemic Abs with cross-reactivity to SARS-CoV-2 nucleocapsid (NP) and spike proteins did not protect against severe symptoms [6]. Humoral immunity to hCoVs (NL63 and 229E [7]), respiratory syncytial virus, cytomegalovirus and herpes simplex virus-1 [8, 9] has been associated with more severe COVID-19 disease.

The presence of Abs with affinity for a 21-mer peptide derived from SARS-CoV-2 NP, an epitope region termed Ep9, has been correlated with severe COVID-19. The patients, termed αEp9(+), comprised ≈27% of the sampled SARS-CoV-2-infected population (n = 186). The αEp9(+) patients (n = 46) had high, early levels of αN IgGs, typically within the first week, compared to αEp9(−) patients; αEp9(+) individuals also experienced cytokine-related immune hyperactivity [10]. These two observations suggest an AIN-based mechanism for the disease severity observed in αEp9(+) patients. Here, we explore the epitope homology landscape and αEp9 Ab cross-reactivity to potentially identify a primary antigen driving Ab-based immune response in αEp9(+) patients.

## Results and discussion

Assays measured levels of αEp9 IgGs and IgMs from αEp9(+) patients whose plasma was collected at various times post-symptom onset (PSO). Consistent with the hallmarks of AIN tracing a prior infection, αEp9 IgG levels appeared elevated as early as one day PSO in one patient. Similar IgG levels were observed in the patient population over >4 weeks (one-way ANOVA, p = 0.321); thus, αEp9 IgG started high and remained high. Levels of αEp9 IgMs amongst patients at various times PSO were also similar (one-way ANOVA, p = 0.613). The signals measured for αEp9 IgM levels were significantly lower than the equivalent αEp9 IgG levels (t-test, p = 0.0181) (S1 Fig); this difference could reflect lower IgM affinity, quantity, or both. Since the study focuses on identifying epitope binding by Abs upregulated in SARS-CoV-2 positive patients, we cannot discern between a single Ab or a population of Abs with the same binding profile. Additionally, the observation that the Ep9 epitope is targeted by both IgG and IgM antibodies suggests that multiple antibodies with similar binding profiles may exist in SARS-CoV-2 patients. Therefore, we refer to the anti-Ep9 paratopes as belonging to a population of Abs in sera.

Searches for sequence and structural homologs of Ep9 using pBLAST [11] and VAST [12] databases suggested candidate primary antigens. A structural homolog from betaherpesvirus 6A and 14 other Ep9 sequence homologs were identified. Additionally, Ep9-orthologous regions from six human coronaviruses (SARS-CoV, MERS, OC43, HKU-1, NL63, 229E) were chosen for subsequent assays (Fig 1A and S1 Table). To expedite the binding measurements, the potential AIN epitope regions were subcloned into phagemids encoding the sequences as fusions to the M13 bacteriophage P8 coat protein. DNA sequencing and ELISA experiments demonstrated successful cloning and consistent phage display, respectively. Two epitopes failed to display on phage and were omitted from subsequent investigation (S2 Table and S2A Fig).

Since patient samples were collected at different time points during the patients' infection, Ab levels varied significantly between patients. Thus, patients' samples were pooled for the initial assays to minimize outlier concentrations and best capture the average Ab population in patients. The pooled sample data were first used to screen for cross-reactivity against multiple possible epitopes (Fig 1C and 1F). These results were then re-examined with assays of samples from individual patients (Fig 2A). In these experiments, phage ELISAs tested binding by Ep9

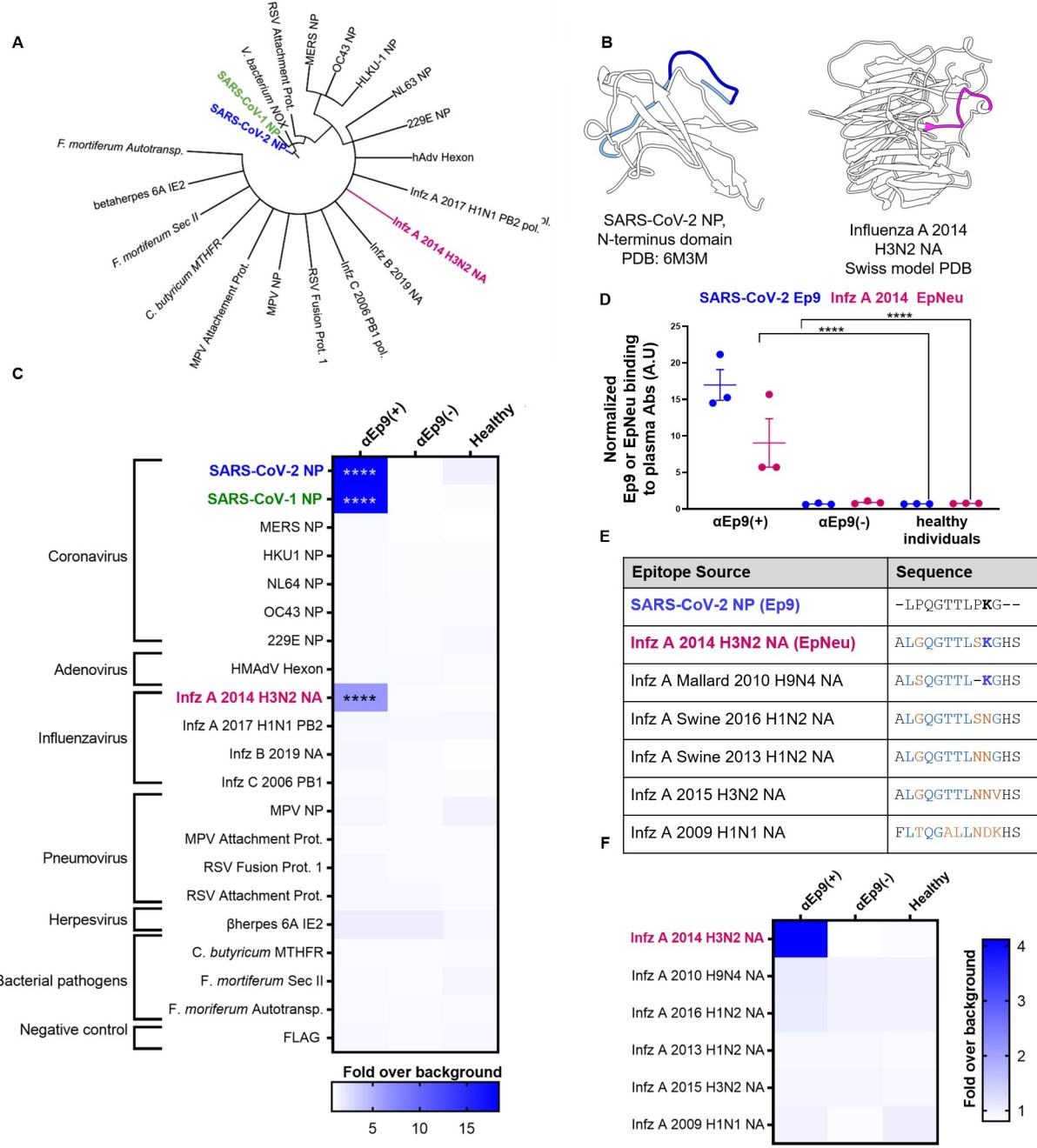

Fig 1. Potential OAS epitopes for binding αEp9 Abs suggested by bioinformatics and tested by phage ELISA. (**A**) Cladogram depicting sequence homology of the Ep9 sequence from SARS-CoV-2 to the bioinformatics-identified, closest homologs. Sequence alignments used pBLAST and VAST, and the cladogram was generated by iTOL [13]. (**B**) Structures of SARS-CoV-2 NP RNA binding domain (PDB: 6M3M) and the influenza virus (Infz) A 2014 H3N2 NA protein (modeled by SWISS-Model [14]). SARS-CoV-2 NP highlights Ep9 residues (light and dark blue) and the region homologous region to EpNeu (dark blue). The depicted model of Infz A 2014 H3N2 NA highlights the EpNeu putative antigen (pink). (**C**) ELISAs examined binding of phage-displayed potential OAS epitopes to total Ig from three sets of pooled plasma from five αEp9(+) patients, or five αEp9(−) patients. Pooled plasma from healthy individuals was an additional negative control. The colors of the heat map represent the mean binding signal normalized to phage background negative controls (signal from phage without a displayed peptide). (**D**) Expansion of data from panel C shows ELISA signals from the independently assayed individual pools shows results from the individual pools (****p <0.0001 for a two-way ANOVA comparing binding of phage-displayed epitopes listed in panel C to different groups of pooled plasma, *ad hoc* Tukey test). (**E**) Amino acid sequence alignment of the closely related Infz A NA homologs of EpNeu from pBLAST [11]. Blue and orange residues represent conserved and mismatched amino acids, respectively, relative to Ep9. Bolded residues are important for epitope recognition by αEp9 Abs. (**F**) Using EpNeu as the search template to generate homologous sequences (shown in panel E), ELISAs

examined EpNeu homologs' binding to pooled plasma from αEp9(+), αEp9(−), or healthy individuals. The data are represented as described in panel C (****p <0.0001 for two-way ANOVA c phage-displayed epitopes, *ad hoc* Tukey and Dunnett's test as shown).

homologs to αEp9 Abs. An average response within the patient population was assessed using pooled plasma from three sets of five αEp9(+) and five αEp9(−) COVID-19 patients coated onto ELISA plates. Plasma from healthy individuals provided an additional negative control. Confirming previously reported results [10], SARS-COV-2 Ep9 and a homologous epitope from SARS-CoV-1 (90% similarity) bound only to plasma from αEp9(+) patients. The αEp9 Ab affinity for SARS-CoV-1 is unlikely to drive SARS-CoV-2 AIN due to the former's limited spread in the US [15].

The panel of potential epitopes revealed a candidate epitope from the neuraminidase (NA) protein of an H3N2 influenza A strain, which circulated in 2014 (A/Para/128982-IEC/2014, Accession No. AIX95025.1), termed EpNeu here. The plasma from three different pools of αEp9(+) patients, but not αEp9(−) patients nor healthy individuals, bound EpNeu (p<0.0001, two-way ANOVA *ad hoc* Tukey test) (Fig 1C and 1D). Additionally, the combined technical replicates from two independent experiments of the same pooled samples also demonstrate significant increases in EpNeu binding signal from αEp9(+) plasma Abs, but not in αEp9(−) patients (EpNeu with p<0.0001, two-way ANOVA *ad hoc* Tukey test) (S1 Fig). Though Ep9 and EpNeu share 38% amino acid sequence similarity, other candidate epitope regions with significantly higher homology failed to bind to αEp9(+) plasma (S1 Table).

Next, the specificity of αEp9 Abs binding to NA from different viral strains was explored. EpNeu provided a template for further homolog searches in sequence databases. Closely aligned NA sequences isolated from human, avian, and swine hosts in North America were chosen for further analysis (Fig 1E, S1 Table). The sequences were phage-displayed as before. Despite their close similarity to EpNeu (up to 92.3% similarity or only one residue difference), none of the EpNeu homologs bound to Abs from αEp9(+) patients (Fig 1F). A single EpNeu amino acid substitution, K142N (numbering from full-length NA, Accession No. AID57909.1) in an H1N2 swine flu (2016) dramatically decreased binding affinity to Abs from αEp9(+) patients (p<0.0001 one-way ANOVA *ad hoc* Tukey). An epitope of H9N4 avian influenza A virus (2010) missing residue S141, but including conserved K142, also greatly reduced binding to Abs from αEp9(+) patients (p<0.0001 one-way ANOVA *ad hoc* Tukey) (Fig 1E and 1F). Therefore, S141 and K142 are critical for binding to αEp9 Abs.

We further examine whether Ep9 and EpNeu epitopes bind the same Abs. Data from 29 αEp9(+) patients demonstrated a strong, highly significant correlation between levels of Abs binding to Ep9 and EpNeu epitopes in patient plasma (Fig 2A and 2B). Cross-reactivity was confirmed by a sandwich-format assay requiring bivalent, simultaneous binding to both eGFP-fused Ep9 and phage-displayed EpNeu (Figs 2C, S4A and S4B). Cross-reactive Ab binding both Ep9 and EpNeu epitopes in pooled plasma from αEp9(+) patients, but not in αEp9(−) patients with other αNP Abs or healthy donors was demonstrated. Thus, we conclude that αEp9 Abs also recognize the EpNeu epitope. The bivalent ELISA was conducted using pooled patient plasma because epitope concentrations coated in wells of the microtiter plate required optimization to adjust for different levels of Abs from each individual patient and to allow bivalent binding to each type of epitope (S4A Fig). Therefore, it was speculated that the average amount of Abs in each pool would be similar and that the repeated optimization would not be required (S4B Fig).

We then investigated whether EpNeu could present a viable antigen during infection with 2014 H3N2 (NCBI: txid1566483). Linear epitope analysis of full-length NA protein (Bepipred 2.0) [16] predicted a candidate antigen with eight residues from EpNeu, including S141 and

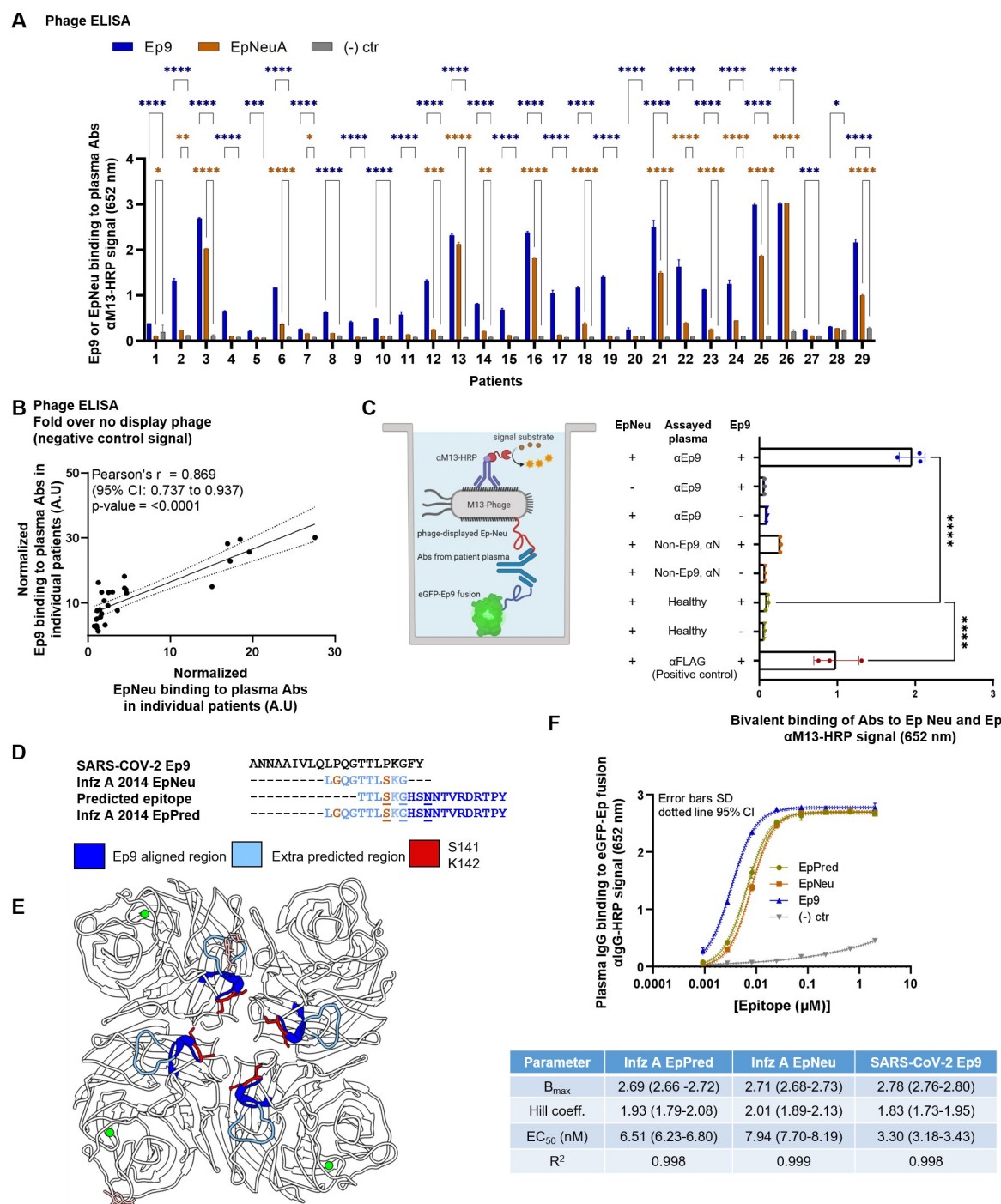

**Fig 2. Cross-reactive Ab binding to both Ep9 and EpNeu, and EpNeu epitope prediction.** (A) Phage ELISA using 29 previously tested αEp9(+) COVID-19 patients. The ELISA demonstrated binding of patient plasma Abs to SARS-CoV-2 epitope, Ep9, or the influenza A neuraminidase epitope, EpNeu. Plasma Abs from 16 out of 29 patients Ep9(+) patients showed significant binding to EpNeu. (****p<0.0001, ***p<0.001, **p<0.01, *p<0.05, two-way ANOVA *ad hoc* Tukey test shown) Significant differences in epitope binding in comparison to the no peptide displayed phage signals are denoted as blue for Ep9 and orange for EpNeu. (B) Comparing normalized levels of phage-displayed Ep9 and EpNeu binding to plasma-coated wells from individual αEp9(+) patients (n = 29). A strong correlation is observed, as shown by the depicted statistics. Each point in panels A through C represents data from individual patients. (C) A schematic diagram of the sandwich ELISA to examine cross-reactivity of αEp9 Abs. The assay tests for bivalent Ab binding to both Ep9 and EpNeu. Pooled plasma from five αEp9(+) patients or five αEp9(−) patients with other αNP Abs was tested for bivalent binding to both eGFP-fused

Ep9 and phage-displayed EpNeu. Healthy patient plasma was used as a negative control. For additional negative controls, phage-FLAG and eGFP-FLAG replaced Ep9 and EpNeu, respectively (****p <0.0001 one-way ANOVA, *ad hoc* Tukey and Dunnett's test shown, with healthy plasma in the presence of EpNeu and Ep9 as negative control). Error bars represent SD. Individual points on bar graph represent technical replicates. (D) Linear and structural B-cell epitope prediction tools Bepipred 2.0 [16] and Discotope 2.0 [17] suggested an extended, linear epitope region from the influenza virus A H3N2 2014 NA, including the eight residues of Ep9 Neu (light blue) with an additional ten, C-terminal residues (dark blue). This extended, predicted epitope is termed EpPred. Structural epitope predictions are underlined. Residues on EpNeu that are not aligned with Ep9 are depicted in orange. (E) Structural model depicting the influenza A H3N2 2014 NA. The model was generated using SWISS-Model based on the NA structure from influenza A H3N2 Tanzania 2010 (PDB: 4GZS). The NA structure highlights the EpNeu region (light blue), the extended residues in EpPred (dark blue), potential glycosylation sites (light pink), and the residues S141 and K142 (red), which are important for αEp9 Ab recognition. (F) Dose-dependent ELISA comparing binding of αEp9 Abs to Ep9, EpNeu and EpPred. Pooled plasma from five αEp9(+) patients and five αEp9(−) patients were tested in triplicates with varying concentrations of eGFP-fused epitopes. The data demonstrates the strongest interactions occurred between αEp9 Abs and Ep9 with an approximately 2-fold decrease in αEp9 Abs binding affinity for EpNeu. EpPred bound slightly stronger to αEp9 Abs than EpNeu; the difference in trend lines of EpNeu and EpPred are statistically significant (p<0.0001, Comparison of Fits). Trend lines represent non-linear regression fit with Hill slope analysis.

K142, and ten additional residues (146–155). This predicted epitope region, termed EpPred, includes the conserved catalytic NA residue D151 targeted for viral neutralization by the immune system [18] (Figs 2D and S5A). A model structure of 2014 H3N2 NA from Swiss-Model [14, 19] and structural epitope prediction (Discotope 2.0) [17] also identified potential epitopes within EpPred (Figs 2D and 2E and S5B).

eGFP-fused EpPred (S2B Fig) was assayed with pooled plasma from five αEp9(+) patients. Controls included EpNeu and Ep9 (positive) and eGFP FLAG (negative). The αEp9 Abs bound to Ep9 with ≈2-fold stronger apparent affinity than for EpNeu (Fig 2E). The increased binding strength of Ep9 could result from additional rounds of Ab affinity maturation after the primary infection [3]. The longer length EpPred appears to modestly improve upon binding of EpNeu to αEp9 Abs (Fig 2F). Thus, while αEp9 Abs may target a larger epitope of H3N2 2014 NA beyond regions homologous to Ep9, the known balkiness of full-length NA's to over-expression makes this hypothesis difficult to test [20]. Additionally, the bacterially overexpressed epitopes assayed here do not include post-translational modifications. Taken together, the results are consistent with the hypothesis that αEp9 Abs found in severe COVID-19 can result from AIN with H3N2 influenza A virus.

Unfortunately, patient histories typically do not include influenza infections and vaccinations. Isolated from Para, Brazil, the H3N2 2014 strain has unknown spread in North America. However, a severe outbreak of influenza A was recorded in 2014 [21, 22]. Since only hemagglutinin was sequenced for strain identification in 2014 [22], the candidate AIN strain from the current investigation could not be effectively traced as only its NA sequence was available. Notably, the EpNeu homolog from the 2014 vaccine H3N2 strain (identical to influenza A 2015 H3N2 NA, Accession No. ANM97445.1) does not bind αEp9 Abs (Fig 1E and 1F). Therefore, αEpNeu Abs must have been generated against a primary influenza virus infection, not the vaccine.

Next, we analyzed Ep9 and EpNeu binding by αEp9 IgGs relative to days PSO (S6A Fig). Cross reactive αEp9 IgGs were observed within one day PSO. The observation is consistent with the imprinting hypothesis, whereby mature IgGs from a previous infection would be present early in the course of the infection. Though low levels of early αEp9 IgGs bound without EpNeu cross reactivity were observed in one patient at one day PSO, this observation could result from EpNeu binding below the level of detection; αEp9 Ab binds at lower affinities to EpNeu, for example.

Analysis of αEp9 IgG cross reactivity and disease severity demonstrated that cross reactive antibodies were observed in patients presenting with all levels of severity (asymptomatic, outpatient, inpatient, ICU admittance, or deceased) (S6B Fig). While EpNeu binding in most patients was drastically lower than binding to Ep9, a subset of hospitalized or ICU admitted

patients demonstrated αEp9 Abs binding to EpNeu and Ep9 at comparable levels (>50%). Such similar Ab binding levels to both Ep9 and EpNeu are not observed in patients with less severe outcomes (i.e., patients who were asymptomatic or experienced only outpatient visits). However, 86% of the samples tested in this study were from hospitalized and admitted to the ICU patients. Similar levels of Ab binding to both Ep9 and EpNeu in the subset of hospitalized and ICU-admitted patients could suggest impaired affinity maturation in patients with more severe outcomes. Impaired Ab affinity maturation have been previously shown to correlate with COVID-19 severity [23, 24]. While multiple factors may lead to disease severity during COVID-19, our results suggest that a reliance on high levels of imprinted influenza Abs by a subset of COVID-19 patients could be indicative of a less effective immune response and consequently more severe disease outcomes.

This report suggests a possible molecular mechanism for AIN underlying the high-rate of severe COVID-19 in αEp9(+) patients. Specifically, we demonstrate cross-reactive binding between αEp9 Abs and a predicted NA epitope from a 2014 influenza A virus strain. Future studies could examine correlation between a country's rate of the H3N2 2014 influenza virus and severe COVID-19. Additionally, correlation could be tested using health systems that record influenza infections. Examining epitope conservation and Ab cross-reactivity could predict AIN-based immune responses and disease outcomes in future infections. Identifying detrimental, benign, or beneficial AIN pathways could also guide vaccine design.

## Materials and methods

Patient samples were collected by the UC Irvine Experimental Tissue Resource, which operates under a blanket UCI IRB protocol (UCI #2012–8716). Written consent from sample donors was obtained.

### Sequence and structural alignment analysis

To identify possible sources of primary infection responsible for αEp9 Ab generation, sequence and structural alignment with Ep9 residues and the SARS-CoV-2 NP was conducted. Alignment of Ep9 sequence with the orthologs from other human coronaviruses (hCOVs) such as SARS-CoV, MERS, HKU-1, NL63, 229E and OC43 was conducted using the Benchling sequence alignment tool [25] (https://benchling.com). To explore a wider range of human host pathogens pBLAST [11] (https://blast.ncbi.nlm.nih.gov/Blast.cgi) was used to search for Ep9 homology in a database of non-redundant protein sequences; common human-host viruses were specified in the organism category. The queries were conducted with the blastp (protein-protein BLAST) program [11] with search parameters automatically adjusted for short input sequences. Alignments spanning >7 residues were included here. The Vector Alignment Search Tool (VAST) [12] (https://structure.ncbi.nlm.nih.gov/Structure/VAST/vast.shtml) was used to find structural alignment between SARS-CoV-2 Ep9 and proteins from other viral and bacterial human host pathogens. Alignment for NP from common hCoV were not further examined, as they had been included in sequence alignment analysis. The aligned sequences were sorted by the number of aligned residues as well as root-mean square deviation (RMDS). The top 50 structurally aligned proteins were then examined for structural homology in the Ep9 epitope region. Regions of proteins that aligned with the Ep9 region were selected for subsequent analysis.

### Cloning

Predicted AIN epitopes were subcloned for phage display using the pM1165a phagemid vector [26] with an N-terminal FLAG-tag and a C-terminal P8 M13-bacteriophage coat protein. AIN constructs were subcloned using the Q5 site-directed mutagenesis kit (New England Biolabs,

Ipswich, MA) as per manufacturer's instructions. The above-mentioned cloned phagemids were then transformed into XL-1 Blue *E. coli* and spread on carbenicillin-supplemented (50 μg/ml) plates. Individual colonies of were then inoculated in 5 ml cultures, and shaken overnight at 37°C. The phagemid was isolated using the QIAprep spin miniprep kit (Qiagen, Germantown, MD) as per manufacturer's instructions. Cloned sequences were verified by Sanger sequencing (Genewiz, San Diego, CA).

## Phage propagation and purification

The Ep9 homologs were expressed as N-terminal fusions to the P8 coat protein of M13 bacteriophage. Plasmids were transformed into SS320 *E. coli* and spread onto carbenicillin-supplemented (50 μg/ml) LB-agar plates before overnight incubation at 37°C. A single colony was inoculated into a primary culture of 15 ml of 2YT supplemented with 50 μg/ml carbenicillin and 2.5 μg/ml of tetracycline and incubated at 37°C with shaking at 225 rpm until an optical density at 600 nm ($OD_{600}$) of 0.5 to 0.7 was reached. 30 μM isopropyl β-D-1-thiogalactopyranoside IPTG and M13KO7 helper phage at a multiplicity of infection (MOI) of 4.6 was added to the primary culture, and the culture was incubated for an additional 37°C with shaking at 225 rpm for 45 min. 8 ml of the primary culture was then transferred to 300 ml of 2YT supplemented with 50 μg/ml of carbenicillin and 20 μg/ml of kanamycin. The cultures were inoculated at 30°C with shaking at 225 rpm for around 19 h.

The phage propagation culture was centrifuged at 9,632 x *g* for 10 min at 4°C. The supernatant, containing the phage, was transferred into a separate tubes pre-aliquoted with 20% tube volume of phage precipitation buffer (20% w/v PEG-8,000 and 2.5 M NaCl), and incubated on ice for 30 min. The solution, containing precipitated phage, was centrifuged for 15 min at 4°C, and the supernatant was discarded. The precipitated phage was centrifuged a second time at 1,541 x *g* for 4 min at 4°C, and then dissolved in 20 ml of resuspension buffer (10 mM phosphate, 137 mM NaCl, pH 7.4–8.0 with Tween-20 0.05% v/v and glycerol 10% v/v). The resuspended pellet solution was divided into 1 ml aliquots, which were flash frozen with liquid nitrogen for storage in −80°C. Prior to use in ELISAs, the aliquoted phage-displayed constructs were re-precipitated in 0.2 ml of phage precipitation buffer after incubation for 30 min on ice. Aliquots were centrifuged at 12,298 x *g* for 20 min at 4°C and the supernatant was discarded. The phage pellets were re-centrifuged at 1,968 x *g* for 4 min at 4°C, and then resuspended in 1 ml of 10 mM phosphate, 137 mM NaCl, pH 7.4.

## Expression and Purification of eGFP fusion peptides

pET28c plasmids encoding eGFP fusions to C-terminal Ep9-FLAG, EpNeu-FLAG, EpPred-FLAG and FLAG (negative control) and N-terminal $His_6$ peptide epitopes, were transformed into BL21DE3 Star *E. coli* chemically competent cells. Transformants were spread on carbenicillin-supplemented (50 μg/ml) LB-agar plates and incubated at 37°C overnight. Single colonies of each construct were selected to inoculate 25 ml LB media supplemented with carbenicillin (50 μg/ml). After incubation at 37°C with shaking at 255 rpm overnight, 5 ml of seed cultures were used to inoculate 500 ml of LB media supplemented with carbenicillin (50 μg/ml). Expression cultures were incubated at 37°C with shaking at 225 rpm until an $OD_{600}$ of ~0.5 was reached. The cultures were induced with 0.5 mM IPTG and incubated at 25°C for 18 h. The cells were pelleted by centrifugation at 9,632 x *g* for 20 min and resuspended in Tris-HCl lysis buffer (20 mM Tris-HCl, 250 mM NaCl, pH 8). Cells were lysed by sonication and the insoluble fractions were pelleted by centrifugation at 24,696 x *g*. The supernatant was affinity-purified using Profinity™ IMAC (BioRad, Hercules, CA) resin charged with nickel sulfate. The protein lysate was batch bound overnight to the IMAC resin and purified using

gravity columns. Columns were washed with lysis buffer supplemented with 20 mM imidazole, and the elution fractions were collected from lysis buffer containing 250 mM imidazole. The elution fractions were then buffer-exchanged with lysis buffer lacking imidazole using Vivaspin® 20 Ultrafiltration Units (Sartorius, Goettingen, Germany) with a molecular weight cut-off of 10 kDa. The final buffer imidazole concentrations were calculated to be ~0.1 mM. Purified and buffer-exchanged protein fractions were then visualized using 10% sodium dode-cyl sulfate polyacrylamide gel electrophoresis (SDS-PAGE) with Coomassie dye staining.

**Patient sample collection.** Samples were collected as previously described [10]. Briefly, the UC Irvine Experimental Tissue Resource (ETR) operates under a blanket IRB protocol (UCI #2012–8716) which enables sample collection in excess of requirements for clinical diagnosis and allows distribution to investigators. Plasma was collected from daily blood draws of COVID(+) patients, initially confirmed with pharyngeal swabs. After immediate centrifugation, plasma from heparin-anticoagulated blood was stored for 3–4 days at 4°C prior to its release for research use. Personal health information was omitted and unique de-identifier codes were assigned to patients to comply with the Non-Human Subjects Determination exemption from the UCI IRB. At the research facility, SARS-CoV-2 virus in plasma samples was inactivated through treatment by incubation in a 56°C water bath for 30 min [27] prior to storage at −80°C.

## Phage ELISAs

As described in previous reports [10], pooled plasma from five αEp9(+) patients, five αEp9(−) patients, or healthy individuals (Sigma-Aldrich, Saint Louis, MO) were separately prepared in coating buffer (50 mM $Na_2CO_3$, pH 9.6); the plasma was diluted 100-fold during this step. Plasma samples were then immobilized in 96 well microtiter plates by shaking the plasma solutions at 150 rpm at room temperature (RT) for 30 min. After aspiration and washing by plate washer (BioTek, Winooski, VT), each well was blocked with 100 μL of ChonBlock Blocking Buffer (CBB) (Chondrex, Inc., Woodinville, WA) for 30 mins, shaking at 150 rpm at RT. Wells were subsequently washed three times with PBS-T (0.05% v/v Tween-20 in PBS). Next, 1 nM phage-displayed candidate "primary" epitopes and controls prepared in CBB was incubated in microtiter wells for 1 h at RT with shaking at 150 rpm. Unbound phage were aspirated and removed using three washes with PBS-T. The peroxidase-conjugated detection antibody, αM13-HRP (Creative Diagnostics, Shirley, NY), was diluted 1,000-fold in Chonblock Secondary Antibody Dilution (Chondrex, Inc., Woodinville, WA) buffer; 100 μl of this solution was added to each well before incubation for 30 min at RT with shaking at 150 rpm. Following aspiration and three washes (100 μl each), 1-Step Ultra 3,3′,5,5′-Tetramethylbenzidine TMB-ELISA Sub-strate Solution (ThermoScientific, Carlsbad, CA) was added (100 μl per well). Absorbance of TMB substrate was measured twice at 652 nm by UV-Vis plate reader (BioTek Winooski, VT) after 5 and 15 min of incubation. The measurement at 5 mins ensured that ELISAs with strong signals were quantified before oversaturation, and the second measurement at 15 mins was collected to enhance any wells with lower signal levels; the approach ensures that no comparable signals were observed in the negative controls. In ELISAs without oversaturated signals, the measurements at 15 mins were used for data analysis. The experiment was repeated three times using plasma from different αEp9(+) and αEp9(−) patients for each experiments, using a total of 15 patients for each group. Each experiment was conducted in technical duplicate.

## αEp9 IgG and IgM ELISA

Plasma from 46 patients, previously tested for the presence of αEp9 Abs using phage ELISAs [10], were used to test levels of αEp9 IgGs and IgMs. Due to low sample availability, the plasmas from 29 patients were then used to compare IgG binding to the Ep9 and the EpNeu

epitopes. 2 μM eGFP-Ep9 or eGFP-FLAG in PBS pH 8.0 were immobilized onto 96 well microtiter plates via overnight incubation with shaking at 150 rpm at 4˚C. Excess protein was aspirated and removed with three consecutive PBS-T washes. Wells were blocked by adding CBB (100 μl) before incubation at 30 min at RT with shaking at 150 rpm. Next, αEp9(+) patient plasma, diluted 1:100 in CBB (100 μl), was added to duplicate wells before incubation at RT for 1 h with shaking at 150 rpm. The solutions were discarded and sample wells were washed with PBS-T three times. αEp9 Abs binding to the potential epitopes was detected using horse radish peroxidase (HRP) conjugated αHuman Fc IgG (Thermo Fisher Scientific, Waltham MA) or αIgM μ-chain specific (Millipore Sigma, Temecula, CA) Abs diluted 1:5,000 in ChonBlock Sample Antibody Dilution buffer. 100 μl of detection Abs were added to each sample well, and incubated for 30 min at RT with shaking at 150 rpm. Sample wells were aspirated and washed three times in PBS-T, and the binding signal was detected after addition of TMB substrate (100 μl per well).

## Bivalent Abs binding ELISA

eGFP-Ep9 or eGFP-FLAG was serially diluted (120 nM, 40 nM, 13 nM and 4 nM) in PBS pH 8.0, and added to the appropriate wells in 96 well microtiter plates, followed by shaking overnight at 150 rpm at 4˚C. Excess unbound protein was removed, and the plate was washed three times in PBS-T. Wells were then blocked in CBB and incubated for 30 min at RT. After blocking, pooled plasma (100 μl per well) from either five αEp9(+) patients, or five non-αEp9, αNP(+) patients, or healthy individuals was added to the appropriate wells. Plasma from pooled patients was diluted 100-fold in CBB. As a positive control αFLAG Ab was used as a 1:2,000 dilution in CBB. Samples were incubated for 1 h at RT with 150 rpm shaking. The solution was removed by aspiration, and the plate and washed three times with PBS-T. Then 1 nM EpNeu displaying phage or the phage negative control with no epitopes displayed was diluted in CBB. 100 μl phage solution was added to microtiter wells and incubated for 30 min at RT with shaking at 150 rpm. After aspirating and washing off unbound phage, binding of phage-displayed EpNeu to plasma αEp9 Abs was visualized using αM13-HRP Ab diluted 1:10,000 in ChonBlock Sample Antibody Dilution buffer. Samples were incubated for 30 min at RT with 150 rpm shaking, and unbound Abs were removed through washing with PBS-T three times before addition of TMB substrate (100 μl). Experiments were conducted in technical triplicates and repeated three times with different αEp(+) and αEp(−) patient samples.

## Dose-dependent ELISA

Wells of microtiter plates were coated with serially diluted concentration of eGFP-Ep9, EpNeu and EpPred or eGFP-FLAG, and incubated overnight at 4˚C before blocking as described above. Next, pooled plasma (100 μl per well) from either five αEp9(+) patients, or five αEp9(−) patients, or healthy individuals at 1:100 total plasma dilution in CBB was added to the appropriate wells. Samples were incubated for 1 h at RT with shaking at 150 rpm. After incubation, unbound solution was removed, and the plates were washed three times with PBS-T. αEp9 IgG levels were detected by adding αFc IgG-HRP diluted 1:5,000 in ChonBlock Sample Dilution buffer, followed by incubation for 30 min at RT with shaking at 150 rpm, followed by addition of TMB substrate (100 μl per well). Experiments were conducted in technical triplicates and repeated three times with different αEp(+) and αEp(−) patient samples.

## Linear B-cell epitope prediction

Linear epitopes from the Influenza A/Para/128982-IEC/2014(H3N2) neuraminidase protein were predicted using the partial sequence with Accession AIX95025.1 from the National

Center for Biotechnology Information's GenBank and the linear B-cell epitope prediction tool, Bepipred 2.0 [16] (http://www.cbs.dtu.dk/services/BepiPred-2.0/). The prediction thresholds were set to 0.5. The specificity and sensitivity of epitope prediction at this threshold is 0.572 and 0.586, respectively.

## Structure-based B-cell epitope prediction

The structure of Influenza A/Para/128982-IEC/2014(H3N2) neuraminidase protein was modelled using Swiss-Model [14, 19, 28–31] (https://swissmodel.expasy.org/interactive). Using the ProMod3 3.2.0 tool [19], a structural model was generated based on the crystal structure (2.35Å, PDB 4GZS 1.A) of a homologous H3N2 neuraminidase with 96.39% sequence identity. Modelling methods and quality assessments are further detailed in the report below.

The structural model of Influenza A/Para/128982-IEC/2014(H3N2) neuraminidase was used to predict structure-based epitopes. Using the *in silico* online platform Discotope 2.0 [17] (http://www.cbs.dtu.dk/services/DiscoTope-2.0/), structure-based epitope propensity scores were calculated to predict likely B-cell epitope residues. The score of −3.7 was set as the threshold for epitope prediction, which estimates a specificity and sensitivity of 0.75 and 0.47, respectively (S5 Fig).

## Statistical analysis

The ELISA data were analyzed in GraphPad Prism 9 (https://www.graphpad.com). Since the ELISA assays of 21 potential AIN epitopes were conducted over several microtiter plates for repeated experiments, the raw absorbance values for every patient sample were normalized and represented as the ratio of phage negative control to the signal. For heatmaps, two-way Analysis of variance (ANOVA) with a Tukey adjustment for multiple comparisons tests were conducted for the entire dataset of epitopes. For column comparisons of two groups, for example IgM levels and IgG levels in the αEp(+) patients, unpaired, two-tailed, parametric t-tests were applied. Additionally, for column comparisons between more than two groups, for example IgM or IgG levels groups by weeks PSO, One-way ANOVA with a Tukey adjustment for multiple comparisons tests were used. Where indicated, an ANOVA with a Dunnett's adjustment were performed to compare results to healthy Abs interactions to αEp9(+) patient results. Graphs represent SD error bars for technical replicates, defined as replicates of the same conditions in multiple wells of the same plate. Whereas error bars are shown as SEM when an experiment is repeated with different patient sample sets. Correlations between Ep9 and EpNeu levels in patients were determined by plotting normalized values on an XY graph and performing a linear Pearson's correlation coefficient test, where a $r$ coefficient between 1.0–0.7 were considered strong correlations, values between 0.7 and 0.5 were considered a moderate correlation, and values below 0.5 were considered a weak correlation [32]. The significance of the correlation was evaluated based on p-value $<0.05$.

## Supporting information

**S1 File. SWISS-MODEL of 2014 H3N2 neuraminidase.**
(PDF)

**S1 Table. Potential primary epitopes targeted by αEp9 Abs.**
(PDF)

**S2 Table. Primers used to subclone potential original epitopes.**
(PDF)

**S1 Fig. Early upregulation of αEp9 IgGs.** ELISA of αEp9 (**A**) IgG and (**B**) IgM levels in αEp9 (+) patients (n = 34) from plasma collected at the indicated time periods post-symptom onset (PSO). Statistical analysis was conducted using one-way ANOVA, *ad hoc* Tukey test. Error bars represent SEM. (**C**) ELISA results of αEp9 IgG and IgM levels of each αEp9(+) patient displayed relative to patient age. Pearson's correlation coefficient, r, and the 95% confidence intervals depicted as dotted lines, demonstrated no correlation between age and αEp9 Ab levels.
(TIF)

**S2 Fig. Expression of phage-displayed and eGFP-fused potential AIN epitopes.** (**A**) ELISA demonstrating the display of N-terminal FLAG-tagged potential epitopes fused to the N-terminus of the P8 coat protein. Immobilized αFLAG Abs in microtiter wells bind the displayed FLAG-tag and epitope, and binding is detected with αM-13-HRP Abs as usual. Phage with no epitope displayed provide the negative control. Epitopes for mastadenovirus protein (mAdV) P8 and *V. bacterium* NADH oxidoreductase (NOX) did not display on the phage surface. Error bars represent SD values. (**B**) 10% SDS-PAGE gel stained with Coomassie Blue shows His-tag affinity-purified and buffer-exchanged eGFP-fused epitopes, EpPred, EpNeu, FLAG negative control and Ep9.
(TIF)

**S3 Fig. Experimental repeatability of Phage ELISA showing EpNeu binding of αEp9(+) patient plasma Abs.** (**A**) Data showing the repeatability of ELISAs examining binding of phage-displayed EpNeu within a single set of pooled plasma from five αEp9(+) patients, or five αEp9(−) patients. The pooled plasma from healthy individuals was an additional negative control. Each experiment was conducted in duplicate as shown by the error bars (SD). Dots represent the actual signal from each individual ELISA well. Experiment #1 represents the same data shown in Fig 1D, with further details of duplicates. Experiment #2, represents the same pools of patients as experiment #1 but in a separate independent experimental replicate. (**B**) Two-way ANOVA *ad hoc* Tukey test was conducted where all the technical replicates from the two experiments were grouped together and compared. Significant differences are denoted by an asterisk (*) and the corresponding p-values are shown. Plasma Abs binding to EpNeu is significantly higher in the Ep9(+) patient pool compared to the Ep9(-) and healthy patient pools.
(TIF)

**S4 Fig. Optimization of assay to determine cross-reactivity of αEp9 Ab to Ep9 and EpNeu.** (**A**) Sandwich ELISA testing the binding of Abs from the pooled plasma of five αEp9(+) patients, five αEp9(-) patients with other αNP Abs and healthy individuals. This experiment examines bivalent binding to various doses of immobilized eGFP-fused Ep9 epitope (concentrations of 120, 40, 13 or 4 nM) and phage-displayed EpNeu in solution. The data shows that Abs from αEp9(+) patients, but not αEp9(−) or healthy individuals, bivalently bind both EpNeu and Ep9. The positive control (αFLAG 1:2000 fold dilution) at 100 nM eGFP demonstrates concentrations appropriate for bivalent binding to immobilized and in-solution tags. The schematic diagram illustrates the binding observed for bivalence in αEp9 Abs, where the antibody bridges plate-bound eGFP at its high concentrations. Therefore, Fig 2 in the main text uses 4 nM of eGFP Ep9 coated on the plate, and the FLAG positive control uses eGFP at 100 nM. Error bars represent SD. (**B**) A second set pooled plasma from five different αEp9(+) or αEp9(−) patients were tested for bivalent Ab binding. This pool was only surveyed at one dose, in which 2 nM eGFP Ep9 was coated on the ELISA plate. Bivalent binding of αEp9 Abs both Ep9 and EpNeu was exclusively observed in this αEp9(+) patient pool over background levels. As shown in panel (**A**) dose-dependence, the αFLAG positive control binds poorly at

this concentration of coated Ep9.
(TIF)

**S5 Fig. Linear and structural epitope mapping prediction of Influenza A H3N2 neuraminidase.** (**A**) Linear epitope mapping prediction of the Influenza A 2014 H3N2 using Bepipred 2.0[16] demonstrates high prediction scores in a region spanning 18 residues, which includes eight residues from EpNeu (underlined). The additional ten predicted residues were included as part of an extended epitope termed EpPred. (**B**) Structural epitope mapping, using Discotope 2.0[17], of the modelled neuraminidase protein from Influenza A 2014 H3N2 (SWISS-model[14],(3 predicts an epitope of five residues. These were captured by EpPred, including three found in EpNeu.
(TIF)

**S6 Fig. EpNeu-and Ep9-specific binding by plasma Abs relative to days PSO and disease severity.** (**A**) Phage ELISA using 29 previously tested αEp9(+) COVID-19 patients and each sample's days PSO. The ELISA depicts binding of patient plasma Abs to SARS-CoV-2 epitope, Ep9 (blue), or the influenza A neuraminidase epitope, EpNeu (orange). The data is normalized by fold over binding by phage with no displayed epitopes. (B) Normalized levels of phage-displayed Ep9 and EpNeu binding to plasma-coated wells from individual αEp9(+) patients (n = 26) relative to disease severity (asymptomatic, outpatient, inpatient, ICU, and deceased).
(TIF)

# Acknowledgments

We thank Professor Elizabeth Bess and Professor Stacey Schultz-Cherry for helpful conversations, Dr. Kristin Gabriel for initial bioinformatics analysis, and the patients who generously donated their plasma samples (IRB# 2012–8716).

# Author Contributions

**Conceptualization:** Sanjana R. Sen, Gregory A. Weiss.

**Data curation:** Sanjana R. Sen, Emily C. Sanders.

**Formal analysis:** Sanjana R. Sen, Emily C. Sanders, Joni L. Ricks-Oddie, Gregory A. Weiss.

**Funding acquisition:** Gregory A. Weiss.

**Investigation:** Sanjana R. Sen, Emily C. Sanders, Alicia M. Santos, Keertna Bhuvan, Derek Y. Tang, Aidan A. Gelston, Brian M. Miller.

**Methodology:** Sanjana R. Sen, Emily C. Sanders, Alicia M. Santos, Keertna Bhuvan, Derek Y. Tang, Aidan A. Gelston, Brian M. Miller, Gregory A. Weiss.

**Project administration:** Gregory A. Weiss.

**Resources:** Gregory A. Weiss.

**Supervision:** Sanjana R. Sen, Emily C. Sanders, Gregory A. Weiss.

**Validation:** Sanjana R. Sen, Emily C. Sanders, Alicia M. Santos, Keertna Bhuvan, Derek Y. Tang, Aidan A. Gelston, Brian M. Miller, Gregory A. Weiss.

**Visualization:** Sanjana R. Sen, Emily C. Sanders, Alicia M. Santos, Keertna Bhuvan, Derek Y. Tang, Aidan A. Gelston.

**Writing – original draft:** Sanjana R. Sen, Emily C. Sanders, Derek Y. Tang, Aidan A. Gelston, Gregory A. Weiss.

**Writing – review & editing:** Sanjana R. Sen, Joni L. Ricks-Oddie, Gregory A. Weiss.

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
