## [Decision Letter · Decision Letter 0]

10 Feb 2022

PONE-D-21-37786Evidence for Deleterious Effects of Immunological History in SARS-CoV-2PLOS ONE

Dear Dr. Weiss,

Thank you for submitting your manuscript to PLOS ONE. After careful consideration, we feel that it has merit but does not fully meet PLOS ONE’s publication criteria as it currently stands. Therefore, we invite you to submit a revised version of the manuscript that addresses the points raised during the review process. Overall, the reviewers found merit and interest in the hypothesis of the study and its results. However, several reviewers raised similar points regarding use of pooled sera and relatively small sample size. The requested revisions were alternatively listed as "major" or "minor" depending on the reviewer. Please do your best to address each reviewer's concerns, especially for the common points of concern.

We look forward to receiving your revised manuscript.

Kind regards,

Kevin A. Henry

Academic Editor

PLOS ONE

Journal Requirements:

2. Thank you for stating the following in the Acknowledgments/ Funding Section of your manuscript: 

We gratefully acknowledge the support of the UCI COVID-19 Basic, Translational and Clinical Research Fund (CRAFT), the Allergan Foundation, and UCOP Emergency COVID-19 Research Seed Funding. A.M.S. thank the Minority Access to Research Careers (MARC) Program, funded by the NIH (GM-69337). J.L.R.-O. was supported by the National Center for Research Resources and the National Center for Advancing Translational Sciences from the NIH (TR001414).

G.A.W - UCI COVID-19 Basic, Translational and Clinical Research Fund (CRAFT), the Allergan Foundation, and UCOP Emergency COVID-19 Research Seed Funding. A.M.S. - Minority Access to Research Careers (MARC) Program, funded by the NIH (GM-69337, 

https://www.nigms.nih.gov/training/MARC/Pages/USTARAwards.aspx ). J.L.R.-O. was supported by the National Center for Research Resources and the National Center for Advancing Translational Sciences from the NIH (TR001414, https://ncats.nih.gov/funding/ ). The funders had no role in study design, data collection and analysis, decision to publish, or preparation of the manuscript.

Additional Editor Comments:

Several reviewers raised similar points regarding use of pooled sera and relatively small sample size. The requested revisions were alternatively listed as "major" or "minor" depending on the reviewer. Please do your best to address each reviewer's concerns, especially for the common points of concern.

Reviewers' comments:

Reviewer's Responses to Questions

**Comments to the Author**

1. Is the manuscript technically sound, and do the data support the conclusions?

Reviewer #1: Yes

Reviewer #2: No

Reviewer #3: Yes

2. Has the statistical analysis been performed appropriately and rigorously? 

Reviewer #1: Yes

Reviewer #2: Yes

Reviewer #3: I Don't Know

3. Have the authors made all data underlying the findings in their manuscript fully available?

Reviewer #1: Yes

Reviewer #2: Yes

Reviewer #3: Yes

4. Is the manuscript presented in an intelligible fashion and written in standard English?

Reviewer #1: Yes

Reviewer #2: Yes

Reviewer #3: Yes

5. Review Comments to the Author

Reviewer #1: In the current manuscript, Sen et al. provide evidence that antibodies targetting a region of SARS-CoV-2 nucleocapsid protein, previously found to be correlated with COVID-19 severity, can recognize an epitope from the neuraminidase protein of influenza A virus. In light of their results, the authors propose that existing memory from influenza infections, in particular the H3N2 strain that affected he United states in 2014, could trigger a deletereous Ab response that exacerbates COVID-19 severity.

Overall, the authors’ hypotheses and aim is well presented and the results support them. He language is, for the most par, corret, as well as he structure of the manuscript. I have some minro comments

Fig. 1C: Phage binding: Why is plasma directly coated unto plates? Adsorption unto plasic is a highly unspecific process, where all proteins are going to compete for binding to plastic, making any comparison complicated. A beter approach would be to coat the antigen, apply the plasma, and then detect IgG or IgM bound (as the authors do for the antiEp9 IgG and IgM ELISAs). Also, why is the OD measured at different times?

Fig. 1E vs 1F: Fig. F is inroduced earlier into the text, so the panels should be switched. At the same time, the table contains an extra putative epitope not tested in panel E.

Fig. 1, 2: Is there a reason to pool patient samples? Is it due to big differences in their Ab titers? Otherwise, plotting individual samples, even stratified by Ep9 levels, would be more informative.

Fig. 2: A good addition would be to run a competition study to demonstrate that Ep-Neu and Ep9 share the same paratope. Also, the axis naming is no very clear (2A), or descriptive (2B), or consistent (AE, e.g. Bound serum IgG (OD 652 nm) and Epitope concentration (µM)). The minute differences between EpPred and EpNeu should be evaluated with a technique to evaluae (e.g. SPR) to conclude anything (L205–207).

Minor comments:

L78, 83, 85, 273: Incorrect placement of commas, such as “hCoVs, NL63 and 229”, “comprised 27% of the sampled, SARS-CoV-2-infected population”, “cytokine-related, immune hyperactivity”, or “0.05%, v/v”.

L81-82: “The presence of Abs[…] have”, correct to “has”

L 152: The authors refer to H4N6 avian influenza. Do they mean H9N4?

156: A very conversational one. I would recommend a more classical way to introduce and connect the next batch of experiments.

L268: 1/5TH?

Format: Thousands separator use is not consistent (e.g. L271, 277), incorrect use of hyphens for minus emperature (minus symbol), range (n dash without preceding or trailing spaces)

Reviewer #2: This manuscript investigates whether there is homology between the Ep9 epitope of the SARS-CoV-2 N protein and other proteins. The authors previously showed that individuals infected with SARS-CoV-2 that had Ep9-specific antibodies had a worse prognosis. Here, they investigate whether this could be due to antigenic imprinting and therefore search for cross-reactive epitopes. While this is an interesting hypothesis, the data do not convincingly support it due to the concerns listed below.

• In Figure 1, the data demonstrating binding to the different epitopes is done with 3 sets of pooled serum. While the differences are statistically significant, it is difficult to determine how relevant they are when only n=3 is shown and the experiment is not repeated.

• It would be helpful to see binding to the eGFP-Eph fusion protein by ELISA for each individual person.

• In Figure 2A, they analyze 34 samples independently, and show that only 6/34 of the samples bind to both Eph and EpNeu. Thus, although there is a significant correlation in the values, not all Eph+ individuals are EpNeu+.

• The experiment measuring cross-reactivity between the two antigens by sandwich ELISA shows technical replicates (n=3) of 1 pool of plasma. Therefore, it is not possible to determine if there is binding to both antigens in more than one person.

• The conclusions that cross-reactivity between the epitopes could result in antigenic imprinting are not supported by the data. At best, the data show that there may be cross-reactivity between these similar epitopes. However, the fact that one amino acid substitution in the NP protein of other influenza strains completely blocks binding, rather than a reducing binding raises the question of whether the binding to the EphNue is real. One would also expect more cross-reactivity with that epitope in HKU1 and OC43 as there are only 1-2 amino acid differences between this epitope in these viruses and SARS-CoV-2.

• Finally, if cross-reactivity to EphNeu was causing antigenic imprinting and negatively impacting generation of antibodies specific for SARS-CoV-2 Eph9 in some individuals, one would expect that you would detect EpNeu reactivity in some healthy controls. In other words, if prior exposure to the 2014 influenza was responsible for the variability in SARS-CoV-2 infection, then you would expect to see reactivity in the general population prior to SARS-CoV-2 infection.

Reviewer #3: Sen et. al., report an interesting study examining the hypothesis that some severe Covid-19 cases may be the result of antigenic interference. They propose a mechanism where a pre-existing antibody response to a H3N2 influenza infection, resulted in cross-reactivity to the SARS-COV2 Ep9 epitope. This is an interesting and compelling hypothesis. The authors carry out ELISA based experiments to demonstrate that Ep9 antibody containing plasma cross-reacts with a homologous epitope found in the H3N2 neuraminidase. Having previously established a correlation between Ep9 containing sera and Covid-19 severity, a picture emerges whereby previous infection by H3N2 may explain increased Covid-19 severity in Ep9+ patients.

In general, the study is well executed and appropriate controls are included. The manuscript is well written, and this reviewer views this research favorably.

The main issue with the study is related to the relatively small sample size. For all the ELISA studies, pooled plasma (n= 3) from 5 individuals was used. The effect size is rather pronounced, as the ep9- and negative controls have essentially zero binding. The reviewer lacks the expertise to make the judgment regarding if this sample size is sufficient to draw a definitive conclusion. To this end, either the authors need to increase their sample size, or provide sound reasoning why such a small sample size is sufficient or carefully qualify their results in light of the small sample size.

Some minor points that could use additional clarification:

1. The authors describe an initial plasma collection from 34 individuals, but then make use of pooled plasma for their experiments. Each pool contains plasma from 5 donors. The authors should explain the rationale for using pooled plasma as opposed to plasma from individuals.

2. It is somewhat unclear what the number of healthy donors was for their negative control. Is it also 3 samples of pooled plasma, each with 5 donors?

3. Page 2 lines 57+58. - The statement is also somewhat unclear. Do they mean they discovered a particular antibody or do they mean a population of antibodies in sera (i.e. antibodies?)

6. PLOS authors have the option to publish the peer review history of their article (what does this mean?). If published, this will include your full peer review and any attached files.

Reviewer #1: No

Reviewer #2: No

Reviewer #3: **Yes: **Cory Brooks

---

## [Author Response · Author response to Decision Letter 0]

8 Apr 2022

>We thank the Editor and Reviewers for their time and insights. With this revision, we have responded to all of their suggestions and comments. The resulting manuscript benefits from the changes and is significantly stronger. 

Journal Requirements:

This revision includes changes to the manuscript’s format as required by the guidelines. 

2. Thank you for stating the following in the Acknowledgments/ Funding Section of your manuscript: 

We gratefully acknowledge the support of the UCI COVID-19 Basic, Translational and Clinical Research Fund (CRAFT), the Allergan Foundation, and UCOP Emergency COVID-19 Research Seed Funding. A.M.S. thank the Minority Access to Research Careers (MARC) Program, funded by the NIH (GM-69337). J.L.R.-O. was supported by the National Center for Research Resources and the National Center for Advancing Translational Sciences from the NIH (TR001414).

G.A.W - UCI COVID-19 Basic, Translational and Clinical Research Fund (CRAFT), the Allergan Foundation, and UCOP Emergency COVID-19 Research Seed Funding. A.M.S. - Minority Access to Research Careers (MARC) Program, funded by the NIH (GM-69337, 

https://www.nigms.nih.gov/training/MARC/Pages/USTARAwards.aspx ). J.L.R.-O. was supported by the National Center for Research Resources and the National Center for Advancing Translational Sciences from the NIH (TR001414, https://ncats.nih.gov/funding/ ). The funders had no role in study design, data collection and analysis, decision to publish, or preparation of the manuscript.

>We have removed funding information and confirm that the statement highlighted above is accurate. No further changes will be made. 

>Done.

The corresponding author (Gregory Weiss) is a full-time researcher at UC Irvine, at the affiliated institution. 

Additional Editor Comments:

Several reviewers raised similar points regarding use of pooled sera and relatively small sample size. The requested revisions were alternatively listed as "major" or "minor" depending on the reviewer. Please do your best to address each reviewer's concerns, especially for the common points of concern.

>Thank you very much for giving us an opportunity to respond, which we have done comprehensively as described below.

Reviewers' comments:

Reviewer's Responses to Questions

Comments to the Author

1. Is the manuscript technically sound, and do the data support the conclusions?

Reviewer #1: Yes

Reviewer #2: No

Reviewer #3: Yes

2. Has the statistical analysis been performed appropriately and rigorously?

Reviewer #1: Yes

Reviewer #2: Yes

Reviewer #3: I Don't Know

3. Have the authors made all data underlying the findings in their manuscript fully available?

Reviewer #1: Yes

Reviewer #2: Yes

Reviewer #3: Yes

4. Is the manuscript presented in an intelligible fashion and written in standard English?

Reviewer #1: Yes

Reviewer #2: Yes

Reviewer #3: Yes

5. Review Comments to the Author

Reviewer #1: In the current manuscript, Sen et al. provide evidence that antibodies targetting a region of SARS-CoV-2 nucleocapsid protein, previously found to be correlated with COVID-19 severity, can recognize an epitope from the neuraminidase protein of influenza A virus. In light of their results, the authors propose that existing memory from influenza infections, in particular the H3N2 strain that affected he United states in 2014, could trigger a deletereous Ab response that exacerbates COVID-19 severity.

Overall, the authors’ hypotheses and aim is well presented and the results support them. He language is, for the most par, corret, as well as he structure of the manuscript. I have some minro comments

Fig. 1C: Phage binding: Why is plasma directly coated unto plates? Adsorption unto plasic is a highly unspecific process, where all proteins are going to compete for binding to plastic, making any comparison complicated. A beter approach would be to coat the antigen, apply the plasma, and then detect IgG or IgM bound (as the authors do for the antiEp9 IgG and IgM ELISAs). 

>The reverse phage ELISA method used in this manuscript is standard for the field. See, for example, Chen et al., Science Advances volume 6, number 14, 2020 and Lim et al.,Scientific Reports volume 9, number 6088, 2019. Preliminary experiments testing both reverse and direct ELISA methods demonstrated significantly lower background signals with the reverse phage ELISA approach when used with the commercial blocking agent, Chonblock, and therefore, our investigations were based on this standard method of analysis. 

Also, why is the OD measured at different times? 

>L359-361 states that “Absorbance of TMB substrate was measured twice at 652 nm by UV-Vis plate reader (BioTek Winooski, VT) after 5 and 15 min of incubation.” To clarify this, we have added the following to the manuscript, L361-365 “The measurement at 5 min ensured that ELISAs with strong signals were quantified before oversaturation, and the second measurement at 15 min was collected to enhance any wells with lower level signals; the approach ensures that no comparable signals were observed in the negative controls. In ELISAs without oversaturated signals, the measurements at 15 min were used for data analysis.”

Fig. 1E vs 1F: Fig. F is inroduced earlier into the text, so the panels should be switched. At the same time, the table contains an extra putative epitope not tested in panel E. 

>The figure order has been changed. The extra, untested putative epitope from the table has been removed.

Fig. 1, 2: Is there a reason to pool patient samples? Is it due to big differences in their Ab titers? Otherwise, plotting individual samples, even stratified by Ep9 levels, would be more informative. 

>Once a cross-reactive epitope was identified, individual patients were analyzed for the prevalence of the cross-reactive Ab, and a new figure Fig. 2A has been added to the paper with this data. As suggested by the reviewer, we further expanded the data in figure 2A to demonstrate Ab binding levels to both Ep9 and EpNeu for each individual patient in Fig 1G. 

>This revision also retains the data from pooled patient samples for the following reason (text added to the manuscript). L139-140 “Since patient samples were collected at different time points during the patients’ infection, Ab levels varied significantly between patients. Thus, patients’ samples were pooled for some assays to minimize outlier concentrations and best capture the average Ab population in patients.” 

>Additionally, while we observed anti-Ep9 Ab levels in individual patients, it was unknown whether anti-Ep9 Abs derive from a single type of Ab or a population of Abs targeting the same large epitope. Therefore, pooling Ep9 positive patients (as in Fig. 1 C, D and E) provided an efficient approach for screening potentially slightly variant Abs to determine if any patients within the pool had cross-reactive Abs. 

Fig. 2: A good addition would be to run a competition study to demonstrate that Ep-Neu and Ep9 share the same paratope. Also, the axis naming is no very clear (2A), or descriptive (2B), or consistent (AE, e.g. Bound serum IgG (OD 652 nm) and Epitope concentration (µM)). The minute differences between EpPred and EpNeu should be evaluated with a technique to evaluae (e.g. SPR) to conclude anything (L205–207).

>Fig 2 A and 2E cannot have similar axes as they are two different types of ELISAs. Fig 2A (now 2B) is a phage ELISA, using a specific concentration of epitope-displayed phage and the readout is anti-M13-HRP, normalized to no display controls. Here the values have to be normalized to negative controls for facile comparisons between individual patients. These negative control values and individual patient data have now been included as the new Fig 2A. Fig 2E (now 2F), on the other hand, demonstrates dose-dependent binding of these epitopes for comparison of their EC50s. Therefore, instead of using multivalent phage-displayed epitopes, eGFP-fused epitopes are used. For the latter, the readout is anti-IgG-HRP signal, and the negative control has been shown.

>The positive signal in the bivalent Ab binding in the ELISA on Fig 2B demonstrates that a single antibody binds both epitopes Ep9 and Ep Neu, thereby confirming that they do in fact share the same paratrope. 

>The suggestion to perform the SPR experiment is an interesting one, but unfortunately the quantities of patient samples remaining are too small for this experiment. Since we are analyzing the epitopes removed from the full-length NA protein (due to the known difficulties in overexpression, as explained in the manuscript, L234), the relative binding of the Abs to the different epitopes can be more relevant than the Kd values. Thus, the EC50s from the ELISAs should be sufficient for such comparisons. 

>To address reviewer concerns about the minute differences in binding, we have made changes to the text to more precisely describe the trends observed. In L232-235 we have added “The longer length EpPred appears to modestly improve upon binding of EpNeu to αEp9 Abs (Fig. 2E). Thus, while αEp9 Abs may target a larger epitope of H3N2 2014 NA beyond regions homologous to Ep9, the known balkiness of full-length NA’s to overexpression makes this hypothesis difficult to test[18]”

Minor comments:

L78, 83, 85, 273: Incorrect placement of commas, such as “hCoVs, NL63 and 229”, “comprised 27% of the sampled, SARS-CoV-2-infected population”, “cytokine-related, immune hyperactivity”, or “0.05%, v/v”.

L81-82: “The presence of Abs[…] have”, correct to “has”

L 152: The authors refer to H4N6 avian influenza. Do they mean H9N4?

156: A very conversational one. I would recommend a more classical way to introduce and connect the next batch of experiments.

L268: 1/5TH?

Format: Thousands separator use is not consistent (e.g. L271, 277), incorrect use of hyphens for minus emperature (minus symbol), range (n dash without preceding or trailing spaces)

>The above changes have been made. We thank the reviewer for their excellent suggestions, which have strengthened our manuscript. 

Reviewer #2: This manuscript investigates whether there is homology between the Ep9 epitope of the SARS-CoV-2 N protein and other proteins. The authors previously showed that individuals infected with SARS-CoV-2 that had Ep9-specific antibodies had a worse prognosis. Here, they investigate whether this could be due to antigenic imprinting and therefore search for cross-reactive epitopes. While this is an interesting hypothesis, the data do not convincingly support it due to the concerns listed below.

• In Figure 1, the data demonstrating binding to the different epitopes is done with 3 sets of pooled serum. While the differences are statistically significant, it is difficult to determine how relevant they are when only n=3 is shown and the experiment is not repeated.

>To address the reviewer’s concerns we have added new Fig. S1A, which shows the two independent replicates of a pool of aEp9(+) and aEp9(-) patients, in addition to the pooled plasma from healthy patients (Sigma). The experiment includes three technical replicates for the three different pools of patients (n = 5 patients for aEp9(+) or aEp9(-)). Error bars and data points show data from the technical replicates for each experiment. Additionally, two-way ANOVA ad hoc Tukey of replicates from both experiments show significant increases in EpNeu binding signal from aEp9(+) plasma Abs, but not in aEp9(-) patients. This data has also been described in the manuscript on L155-158. 

• It would be helpful to see binding to the eGFP-Eph fusion protein by ELISA for each individual person.

>Done – in the new Fig. 2A. Plasma Abs from previously confirmed aEp(+) patients were individually tested for Ep9 and EpNeu binding. 

• In Figure 2A, they analyze 34 samples independently, and show that only 6/34 of the samples bind to both Eph and EpNeu. Thus, although there is a significant correlation in the values, not all Eph+ individuals are EpNeu+.

>To clarify, of 34 aEp9(+) patients, 29 were independently tested for anti-EpNeu Abs. A correction was made to L166 and we have added a statement of clarification on L370-372 “Due low sample availability for 5 patients, the plasmas from 29 patients were then used to compare IgG binding to the Ep9 and the EpNeu epitopes.” Of these 29 patient samples, 16 showed significant increase in Ab binding over background. The 6 patients that the reviewer is referring to are the patients that demonstrate the highest levels of binding to both EpNeu and Ep9.

• The experiment measuring cross-reactivity between the two antigens by sandwich ELISA shows technical replicates (n=3) of 1 pool of plasma. Therefore, it is not possible to determine if there is binding to both antigens in more than one person.

>The new Fig. S4B presents additional data wherein plasma from five different aEp9(+) or aEp9(-) patients were pooled and tested for the Ab-mediated bivalent interaction. This data also shows such Ab cross-reactivity. Therefore, it can be concluded that the cross-reactive binding is reproducible amongst the patient population and not due to a single patient. 

• The conclusions that cross-reactivity between the epitopes could result in antigenic imprinting are not supported by the data. At best, the data show that there may be cross-reactivity between these similar epitopes. However, the fact that one amino acid substitution in the NP protein of other influenza strains completely blocks binding, rather than a reducing binding raises the question of whether the binding to the EphNue is real. One would also expect more cross-reactivity with that epitope in HKU1 and OC43 as there are only 1-2 amino acid differences between this epitope in these viruses and SARS-CoV-2.

>Evidence suggests that both broad and narrow antibodies are generated against the influenza virus. In the case of narrow Abs, single site mutations are sufficient for viruses to escape neutralization. This is further investigated in Doud, MB et al. (How single mutations affect viral escape from broad and narrow antibodies to H1 influenza hemagglutinin, Nature Communications, volume 9, article number 1386, 2018).

 Despite there being high homology in EpNeu region between SARS-CoV-2 and HKU1 or OC43, it does not contain the PKG motif but instead has the sequence PQG. As such we observe that Ep9 patients do not have Abs that target this region. This provides further evidence that the single amino acid is important for Ep9 Ab recognition.

• Finally, if cross-reactivity to EphNeu was causing antigenic imprinting and negatively impacting generation of antibodies specific for SARS-CoV-2 Eph9 in some individuals, one would expect that you would detect EpNeu reactivity in some healthy controls. In other words, if prior exposure to the 2014 influenza was responsible for the variability in SARS-CoV-2 infection, then you would expect to see reactivity in the general population prior to SARS-CoV-2 infection.

>As stated on L345, healthy samples were commercially purchased from pooled healthy patients Sigma-Aldrich. As such their medical records or dates of collection were not available. Therefore, it’s hard to know whether these healthy patients would likely have been exposed to the 2014 EpNeu infection. Additionally, the Ep9 Abs in COVID-19 patients are prevalent in detectable amounts in their serum due to an active response against infection; healthy patients who may have been previously exposed to the H3N2 2014 influenza strain and generated anti-EpNeu Abs could still have memory B cells, but may not have detectable levels of Abs in their serum. 

Reviewer #3: Sen et. al., report an interesting study examining the hypothesis that some severe Covid-19 cases may be the result of antigenic interference. They propose a mechanism where a pre-existing antibody response to a H3N2 influenza infection, resulted in cross-reactivity to the SARS-COV2 Ep9 epitope. This is an interesting and compelling hypothesis. The authors carry out ELISA based experiments to demonstrate that Ep9 antibody containing plasma cross-reacts with a homologous epitope found in the H3N2 neuraminidase. Having previously established a correlation between Ep9 containing sera and Covid-19 severity, a picture emerges whereby previous infection by H3N2 may explain increased Covid-19 severity in Ep9+ patients.

In general, the study is well executed and appropriate controls are included. The manuscript is well written, and this reviewer views this research favorably.

>We thank the reviewer for their support.

The main issue with the study is related to the relatively small sample size. For all the ELISA studies, pooled plasma (n= 3) from 5 individuals was used. The effect size is rather pronounced, as the ep9- and negative controls have essentially zero binding. The reviewer lacks the expertise to make the judgment regarding if this sample size is sufficient to draw a definitive conclusion. To this end, either the authors need to increase their sample size, or provide sound reasoning why such a small sample size is sufficient or carefully qualify their results in light of the small sample size.

>The individual data of all patients have now been added to the manuscript as new Fig 2A. These ELISAs demonstrate that patients that have Abs against Ep9 also bind the EpNeu epitope. The pooled sample data are initially used to screen for cross-reactivity against multiple possible epitopes and secondly to support conclusions that are observed from epitope binding observed in individual patients. 

Some minor points that could use additional clarification:

1. The authors describe an initial plasma collection from 34 individuals, but then make use of pooled plasma for their experiments. Each pool contains plasma from 5 donors. The authors should explain the rationale for using pooled plasma as opposed to plasma from individuals.

>The experiments were designed to work with very limited patient samples. The pooled sample data were initially used to screen for cross-reactivity against multiple possible epitopes and secondly to support conclusions that are observed from epitope binding observed in individual patients. See Fig 2A demonstrating Ep9 and EpNeu binding of individual patients. Explanation added to manuscript L142-144. Experiments such as the bivalent ELISA were not possible for individual patients as the epitope concentrations coated to the plate had to be optimized for the levels of Abs in each individual patient to allow for bivalent binding to each type of epitope. For pooled patients,it was speculated that the average amount of Abs in each pool would be similar and that the repeated optimization would not be required. An explanation was added to manuscript in L180-184. 

2. It is somewhat unclear what the number of healthy donors was for their negative control. Is it also 3 samples of pooled plasma, each with 5 donors?

>As stated on L345, healthy samples were commercially purchased pooled from healthy patients Sigma-Aldrich.

3. Page 2 lines 57+58. - The statement is also somewhat unclear. Do they mean they discovered a particular antibody or do they mean a population of antibodies in sera (i.e. antibodies?)

>We have added the following text. “Since the study focuses on identifying epitope binding traits of Abs that are upregulated in SARS-CoV-2 positive patients,we cannot discern between a single Ab or a population of Abs of a certain serotype with the same binding profile. Additionally, we observe that the Ep9 epitope is targeted by IgG and IgM antibodies, suggesting that multiple antibodies with similar binding profiles may exist in SARS-CoV-2 patients. Therefore, we refer to the anti-Ep9 epitopes as a population of Abs in sera” (L100-106). 

>Again, we are grateful to the reviewers for their insights and help strengthening the 

manuscript.

---

## [Decision Letter · Decision Letter 1]

2 May 2022

PONE-D-21-37786R1Evidence for Deleterious Effects of Immunological History in SARS-CoV-2PLOS ONE

Dear Dr. Weiss,

Thank you for submitting your manuscript to PLOS ONE. After careful consideration, we feel that it has merit but does not fully meet PLOS ONE’s publication criteria as it currently stands. Therefore, we invite you to submit a revised version of the manuscript that addresses the points raised during the review process.

 One reviewer still had a few concerns that they believed could be addressed through minor revision (primarily of language) as well as a few other corrections.

We look forward to receiving your revised manuscript.

Kind regards,

Kevin A. Henry

Academic Editor

PLOS ONE

Journal Requirements:

Reviewers' comments:

Reviewer's Responses to Questions

**Comments to the Author**

1. If the authors have adequately addressed your comments raised in a previous round of review and you feel that this manuscript is now acceptable for publication, you may indicate that here to bypass the “Comments to the Author” section, enter your conflict of interest statement in the “Confidential to Editor” section, and submit your "Accept" recommendation.

Reviewer #1: All comments have been addressed

Reviewer #2: (No Response)

Reviewer #3: All comments have been addressed

2. Is the manuscript technically sound, and do the data support the conclusions?

Reviewer #1: Yes

Reviewer #2: Partly

Reviewer #3: Yes

3. Has the statistical analysis been performed appropriately and rigorously? 

Reviewer #1: Yes

Reviewer #2: Yes

Reviewer #3: Yes

4. Have the authors made all data underlying the findings in their manuscript fully available?

Reviewer #1: Yes

Reviewer #2: Yes

Reviewer #3: Yes

5. Is the manuscript presented in an intelligible fashion and written in standard English?

Reviewer #1: Yes

Reviewer #2: Yes

Reviewer #3: Yes

6. Review Comments to the Author

Reviewer #1: (No Response)

Reviewer #2: The authors have addressed many of the concerns raised in the previous review. However, I am still not convinced that antibodies generated by a previous influenza infection are having a negative impact on the immune response to SARS-CoV-2 as there is no evidence that antibodies in these samples bind to an epitope that elicits an antibody response after influenza exposure. Additionally, there is no evidence that antibodies in these samples bind to intact NA protein or epitopes that may be presented by infected cells. Moreover, antibodies that impact the immune response to a pathogen via imprinting, should be detected early after exposure and that information is not presented. Since there may not be sufficient number or volume of samples, the authors could modify their conclusions to indicate that their data are consistent with their overall hypothesis, rather than these data support their hypothesis.

Showing the antibody reactivity in each individual in Fig 2A is very helpful. Since the authors make the point that plasma was collected at different times, which may contribute to the variability in antibody levels, it would be very informative to include the day that the sample was collected in Fig 2A. Since the data are presented in a bar graph, the samples could be arranged by day after symptom onset, rather than patient number. This would enable you to assess whether individuals that had increased levels of Ep9 antibodies early also had antibodies reactive to EpNeu, which would support the imprinting hypothesis. If the EpNeu antibodies are generated by a previous infection and they have an impact on SAR-CoV-2 infection, you would expect to see binding to EpNeu early

Line 237 – the results do not necessarily support the hypothesis as it is still not clear whether this epitope is presented during infection or even in the full-length protein. If the full-length NA can’t be made in bacteria, it can be expressed in other cell types.

Thank you for clarifying that 16/29 patients with antibodies reactive against Ep9 also had antibodies reactive against EpNeu. While this is greater than the 6 individuals that have high EpNeu levels, the fact remains that not all Ep9 antibodies cross-react to EpNeu. This should be considered in the discussion. Is there a stronger correlation with disease severity with EpNeu binding compared to Ep9?

Minor points:

• Fig 1C – Is that IgG, IgM or total Ig?

• I think that lines 231-234 refer to Fig 2F, not Fig 2E as indicated.

Reviewer #3: I am satisfied with the author's responses to all concerns raised by the reviewers. I support acceptance of the manuscript as is.

7. PLOS authors have the option to publish the peer review history of their article (what does this mean?). If published, this will include your full peer review and any attached files.

Reviewer #1: **Yes: **Rafael Bayarri-Olmos

Reviewer #2: No

Reviewer #3: No

---

## [Author Response · Author response to Decision Letter 1]

23 Jun 2022

Please see attached file "Response to Reviewers." Thank you.

---

## [Decision Letter · Decision Letter 2]

14 Jul 2022

Evidence for Deleterious Effects of Immunological History in SARS-CoV-2

PONE-D-21-37786R2

Dear Dr. Weiss,

We’re pleased to inform you that your manuscript has been judged scientifically suitable for publication and will be formally accepted for publication once it meets all outstanding technical requirements.

Kind regards,

Kevin A. Henry

Academic Editor

PLOS ONE

Additional Editor Comments (optional):

Reviewers' comments:

Reviewer's Responses to Questions

**Comments to the Author**

1. If the authors have adequately addressed your comments raised in a previous round of review and you feel that this manuscript is now acceptable for publication, you may indicate that here to bypass the “Comments to the Author” section, enter your conflict of interest statement in the “Confidential to Editor” section, and submit your "Accept" recommendation.

Reviewer #2: All comments have been addressed

2. Is the manuscript technically sound, and do the data support the conclusions?

Reviewer #2: Yes

3. Has the statistical analysis been performed appropriately and rigorously? 

Reviewer #2: Yes

4. Have the authors made all data underlying the findings in their manuscript fully available?

Reviewer #2: Yes

5. Is the manuscript presented in an intelligible fashion and written in standard English?

Reviewer #2: Yes

6. Review Comments to the Author

Reviewer #2: (No Response)

7. PLOS authors have the option to publish the peer review history of their article (what does this mean?). If published, this will include your full peer review and any attached files.

Reviewer #2: No

---

## [Editor Report · Acceptance letter]

11 Aug 2022

PONE-D-21-37786R2 

Evidence for deleterious effects of immunological history in SARS-CoV-2 

Dear Dr. Weiss:

I'm pleased to inform you that your manuscript has been deemed suitable for publication in PLOS ONE. Congratulations! Your manuscript is now with our production department. 

Kind regards, 

on behalf of

Dr. Kevin A. Henry 

Academic Editor

PLOS ONE